# Compact Integration of Hydrogen–Resistant a–InGaZnO and Poly–Si Thin–Film Transistors

**DOI:** 10.3390/mi13060839

**Published:** 2022-05-27

**Authors:** Yunping Wang, Yuheng Zhou, Zhihe Xia, Wei Zhou, Meng Zhang, Fion Sze Yan Yeung, Man Wong, Hoi Sing Kwok, Shengdong Zhang, Lei Lu

**Affiliations:** 1School of Electronic and Computer Engineering, Peking University, Shenzhen 518055, China; wangyunping@pku.edu.cn (Y.W.); 1901213029@pku.edu.cn (Y.Z.); zhangsd@pku.edu.cn (S.Z.); 2State Key Laboratory of Advanced Displays and Optoelectronics and Technologies, Department of Electronic & Computer Engineering, The Hong Kong University of Science and Technology, Hong Kong 999077, China; eezxia@ust.hk (Z.X.); zwxab@alumni.ust.hk (W.Z.); eefion@ust.hk (F.S.Y.Y.); eemwong@ust.hk (M.W.); eekwok@ust.hk (H.S.K.); 3College of Electronic and Information Engineering, Shenzhen University, Shenzhen 518060, China; zhangmeng@connect.ust.hk

**Keywords:** low–temperature poly–Si oxide (LTPO), amorphous indium–gallium–zinc oxide (a–IGZO), thin–film transistors (TFT), diffusion barrier, hydrogen–resistant, nitrous oxide (N2O), fluorination, metal–oxide–on–Si (MOOS)

## Abstract

The low–temperature poly–Si oxide (LTPO) backplane is realized by monolithically integrating low–temperature poly–Si (LTPS) and amorphous oxide semiconductor (AOS) thin–film transistors (TFTs) in the same display backplane. The LTPO–enabled dynamic refreshing rate can significantly reduce the display’s power consumption. However, the essential hydrogenation of LTPS would seriously deteriorate AOS TFTs by increasing the population of channel defects and carriers. Hydrogen (H) diffusion barriers were comparatively investigated to reduce the H content in amorphous indium–gallium–zinc oxide (a–IGZO). Moreover, the intrinsic H–resistance of a–IGZO was impressively enhanced by plasma treatments, such as fluorine and nitrous oxide. Enabled by the suppressed H conflict, a novel AOS/LTPS integration structure was tested by directly stacking the H–resistant a–IGZO on poly–Si TFT, dubbed metal–oxide–on–Si (MOOS). The noticeably shrunken layout footprint could support much higher resolution and pixel density for next–generation displays, especially AR and VR displays. Compared to the conventional LTPO circuits, the more compact MOOS circuits exhibited similar characteristics.

## 1. Introduction

As the amorphous silicon thin–film transistors (TFTs) cannot meet the elevated requirements of active–matrix displays on higher resolution, the lower power consumption, and higher flexibility, the low–temperature fabrication, the ultralow off–state current (I_off_), and other superiorities of AOS TFTs [1,2] have attracted attention in the rapidly growing field of research in in display [3] and flexible electronics [4]. The amorphous indium–gallium–zinc oxide (a–IGZO) TFT [5] is evidence of progress in this field. However, AOS TFTs exhibited relatively lower mobility and poorer stability compared to low–temperature poly–Si (LTPS) TFTs, restricting their applications to high–end displays and other cutting–edge electronic systems. To combine the metrics of both LTPS and AOS, the low–temperature poly–Si oxide (LTPO) technology was tested by monolithically integrating LTPS and AOS TFTs. This technology enables a dynamic refresh rate and, thus, considerably reduces power consumption [6]. Unfortunately, as the hydrogen (H) doping is essential for passivating the dangling bonds in LTPS [7], the AOS TFTs would be severely degraded by these H dopants diffused from the LTPS TFTs, due to the abundant generation of defects and donors in the AOS channels [8]. Thus, it is imperative for the further improvement of LTPO systems to suppress the detrimental H–influence on AOS TFTs [9,10]. 

Although metal films have strong H–blocking capabilities [11], they are not suitable for the hybrid integration structures due to potential problems, such as a short circuit. Previous studies proved that the atomic–layer–deposited (ALD) alumina (AlO_X_) was able to weaken H penetration [12,13,14,15,16]. However, the ALD process itself is rich in H and unsuited to large–area applications. The more compatible H–blocking layer was studied in this work. Furthermore, the H–resistant capability of the AOS channel needs fundamental enhancements. The fluorine (F) plasma treatment has been demonstrated to effectively suppress oxygen–related native defects and enhances the H–resilience of a–IGZO [11,14,17]. Given the corrosive effect of F on common dielectrics, a more moderate pretreatment was developed to realize the H–resistant AOS. Given the suppressed H conflict between LTPS and AOS, a more compact integration architecture of LTPO was proposed in this work for developing advanced displays with higher resolutions and pixels per inch (PPI).

## 2. Development of Hydrogen–Resistant a–IGZO TFTs

As shown in Figure 1a, the a–IGZO TFTs were fabricated in the self–aligned top–gate (SATG) architecture; this procedure followed our previously reported process [11,14]. The H–blocking capabilities of passivation layers (PL) were compared to different passivation–layer materials. The controlled passivation layer was 200–nm–thick silicon oxide (SiO_2_) deposited using the plasma–enhanced chemical vapor deposition (PECVD) with silane (SiH_4_) and nitrous oxide (N_2_O) as the reactive gases. Another passivation layer of 200–nm–thick AlO_X_ was reactively sputtered in the argon–O_2_ mixed ambience; the aluminum (Al) target was used for this procedure. On top of the PECVD–SiO_2_ passivation layer, a 5–nm sputtered Al film was fully oxidized in O_2_ at 200 °C for 30 s, forming the SiO_2_/Oxidized–AlO_X_ bilayer. The hydrogenation treatment of these a–IGZO TFTs was implemented by capping these a–IGZO TFTs with H–rich PECVD silicon nitride (SiN_X_:H) deposited with the SiH_4_ and ammonia (NH_3_). These SATG a–IGZO TFTs with a different passivation layer were characterized with the B1500A Semiconductor Parameter Analyzer semiconductor analyzer, before and after the hydrogenation process.

The drain current (*I*_DS_) versus gate voltage (*V*_GS_) transfer curves of transistors with both a channel width (*W*) and a length (*L*) of 100 μm were measured at a drain voltage (*V*_DS_) of 10.1 V. The threshold voltage (*V*_th_) was extracted as the *V*_GS_ corresponding to an *L*/*W*-normalized *I*_DS_ of 10^−6^ A. In Figure 1b, the *V*_th_ of the SiO_2_–passivated a–IGZO TFT was 0.5 V before the hydrogenation and severely degraded to −28.5 V after the hydrogenation, a result suggesting that abundant H dopants diffuse through the SiO_2_ passivation layer into a–IGZO channel [18]. The TFTs with a sputtered–AlO_X_ passivation layer were short–circuited by the hydrogenation (Figure 1c), while the extracted *V*_th_ of the SiO_2_/AlO_X_–passivated TFT only shifted to −11 V (Figure 1d). The results indicated that the film compactness of sputtered AlO_X_ was even worse than that of PECVD SiO_2_, leading to the heavily H–doped conductive channel. With the addition of only 5–nm thermally–oxidized AlO_X_, the SiO_2_/oxidized–AlO_X_ bilayer exhibited much better H–blocking capability than the SiO_2_ single layer. However, the ultrathin thermally–oxidized AlO_X_ cannot completely block the H diffusion, due to the self–limited nature of Al oxidization. Therefore, it is necessary to elevate the intrinsic H–resistance of AOS TFTs against the residual H dopants penetrating through the bilayer passivation layer. 

The H–resistance of SiO_2_–passivated AOS TFTs was further investigated by additionally treating the a–IGZO islands with N_2_O plasma at 150 °C. In Figure 2a,b, the H–effects were compared on the a–IGZO TFTs with different N_2_O treatment times. As shown in Figure 1b, the untreated a–IGZO TFTs were seriously degraded by hydrogenation. The nearly shorted channel is caused by the numerous H dopants that penetrate the SiO_2_ passivation layer and diffuse into a–IGZO. For the same L of 100 μm, the H–induced Δ*V*_th_ (*V*_th_ before and after the hydrogenation) was significantly decreased to −11.4 V by the 60–s N_2_O treatment, as shown in Figure 2a. This suggests that the H–resistance of the N_2_O–treated a–IGZO channel was effectively strengthened. However, the Δ*V*_th_ of 20–μm–*L* TFT shifted to −12.7 V, and the 5–μm–L channel was still shorted. Such an apparent short–channel effect (SCE) reveals that the channel–carrier concentration is gradually increased by the lateral H diffusion; this evidence is consistent with the perfect H–blocking capability of the top metal gate [19].

When the N_2_O treatment was prolonged to 360 s (Figure 2b), the *V*_th_ of long–channel TFTs exhibited almost no degradation after the hydrogenation, while the Δ*V*_th_ of 5–μm–L TFT only slightly shifted to −3.6 V, revealing a significantly strengthened H–resistance of such strongly oxidized a–IGZO. However, the on–state current (*I*_on_) of such strongly oxidized a–IGZO TFT was degraded after the hydrogenation, especially for the 5–μm–L transistor, a result suggesting a significantly increased source/drain (*S*/*D*) resistance (*R*_SD_). Most plausibly, although the resistivity of strongly oxidized a–IGZO can still be effectively reduced by the argon plasma–induced donor defects [17,20], the additional H dopants suppress these donor defects rather than further supply donors; this process could result in an elevated *R*_SD_. As illustrated in Figure 1a, the *L*_SD_ is even longer for the shorter–*L* transistor, causing an even larger *R*_SD_ and thus seriously limiting the *I*_on_ of 5–μm–L TFT (Figure 2b).

To analyze the influence of N_2_O plasma treatment on a–IGZO, the O1s spectra of IGZO were measured by X–ray photoelectron spectroscopy (XPS). The O1s peak consists of three Gaussian subpeaks centered at 532.4 eV, 531.25 eV, and 530.15 eV, respectively corresponding to the hydroxy (–OH) group, oxygen deficiency, and the oxygen bonded with metal cations [21]. As shown in Figure 3, the N_2_O plasma treatment noticeably suppressed the oxygen deficiency peak in the a–IGZO, while the area percentage of the metal–oxygen (M–O) peak was increased by 22.9%. This result suggests that the N_2_O treatment effectively annihilates the defects in a–IGZO, such as oxygen vacancy (*V*_O_) and –OH. The H dopants in AOS are normally believed to either form –OH groups or occupy the Vo site to form metal-hydrogen (M–H) bonds [21], while the M–H bonds are much more thermally stable than the former [22]. With the Vo defects effectively suppressed by the N_2_O plasma, the less–defective a–IGZO may obtains certain capability of H–resistance or H–blocking, while the residual H effect can hardly be eliminated by the prolonged oxidization.

The fluorination can also effectively passivate the O– and H–related defects in AOS [23,24]; thus, it was combined with the N_2_O plasma in this work. The as–deposited a–IGZO was first immersed in the carbon tetrafluoride (CF_4_) plasma at 150 °C for 600 s to form the fluorinated a–IGZO (a–IGZO:F) and then treated with the N_2_O plasma for 360 s in the same PECVD reactor. As shown in Figure 4a, the key parameters of N_2_O–treated a–IGZO:F TFTs were well maintained after the hydrogenation, including the field–effect mobility (*μ*_FE_), subthreshold slope (*SS*), and on/off ratio (Figure 4b). Even for 5–μm–L TFT, the Δ*V*_th_ only slightly degraded by less than −1.5 V. In addition, the R_SD_ after the hydrogenation of a–IGZO is 185.1 kΩ/sq, and a–IGZO:F is 110.0 kΩ/sq, which is reduced by 40%, a reduction that improves the serious degradation of I_on_. The metal fluorine (M–F) bonds formed during the fluorination process have much higher dissociation energies than the counterpart M–O bond [25], as listed in the Inset. Therefore, the combination of fluorination and oxidization furthest suppresses the native defects in a–IGZO and thus effectively enhances the H–resistant capability of a–IGZO TFTs.

## 3. 3D Integration of IGZO and Poly–Si TFTs

The successful development of H–resistant AOS TFT blazes a new trail for LTPO architecture. Distinct from the conventional side–by–side LTPO structure [26], a 3D integration structure of LTPS and a–IGZO TFTs, dubbed metal–oxide–on–Si (MOOS) architecture, was proposed; the demonstration of a basic inverter circuit supported this proposal. The H–resistant a–IGZO:F TFT was also fabricated in the bottom–gate elevated–metal metal–oxide (EMMO) structure [16] and then integrated with the solid–phase–crystallized (SPC) LTPS TFT [27]. The source/drain (*S*/*D*) activation of n+ a–IGZO and p+ LTPS was implemented with the same O_2_ annealing process at 500 °C for 4 h. Distinct from the existing side–by–side LTPO structure [26], a 3D integration structure of LTPS and a–IGZO TFTs was proposed, and this proposal was supported by the demonstration of an inverter circuit module, dubbed metal–oxide–on–Si (MOOS) architecture. As shown in Figure 5a, the CMOS inverter built in the mainstream planar LTPO structure occupies a large footprint and thus cannot meet the increasing demands on high–resolution high–PPI display. The MOOS invertor exhibited a superior layout efficiency by saving nearly 16% area. However, the compact integration of a–IGZO and LTPS TFTs demands a strong H–immunity of IGZO transistor.

In Figure 6a, the full–swing CMOS inverters were successfully demonstrated by integrating the n–type a–IGZO:F TFT and the p–type poly–Si TFT. In light of the process flow of LTPS TFT, only one more photolithography mask of AOS island is added for realizing a compact CMOS circuit, while the traditional CMOS poly–Si circuits require at least two more masks than PMOS poly–Si circuits. Due to the absence of high–performance p–type AOS, the AOS inverters are normally made of unipolar AOS and thus hardly achieve the full swing, or the number of transistors has to be doubled to realize the pseudo–CMOS circuits [28]. The low–cost CMOS circuits can be achieved by hybrid integrating AOS and LTPS, and the integration density can be further enhanced by using the MOOS structure.

The dynamic performance of the traditional LTPO and the proposed MOOS were further compared using the 19–stage ring oscillator. As shown in Figure 6b, both kinds of ring oscillators worked properly, while the oscillating frequency of the MOOS–structure ring oscillator is 184 kHz, slightly smaller than the 201 kHz of the traditional LTPO counterpart. This may originate from the increased capacitance of crossover areas in the more compact MOOS integration structure, which could be improved using a more sophisticated layout design. This indicates that once the influence on AOS of H species from LTPS is considerably suppressed, a 3D integration of AOS and LTPS can be realized, exhibiting great potentials in advanced displays and other cutting–edge applications, such as virtual reality (VR) and augmented reality (AR).

## 4. Results

In this work, a–IGZO TFTs with different passivation layers were comparatively subjected to hydrogenation, and the results recommended the PECVD SiO_2_/thermally–oxidized AlO_X_ bilayer as the H–blocking candidate. The combination of N_2_O pretreatment and fluorine doping was developed to significantly enhance the intrinsic H–resilience of a–IGZO channel by annihilating the native defects. Enabled by the effectively H–resistant AOS TFT, a novel LTPS/AOS integration architecture was tested by stacking the H–resistant IGZO on poly–Si, dubbed metal–oxide–on–Si (MOOS). Comparable electrical characteristics of basic circuits were achieved using both conventional 2D LTPO and 3D MOOS technologies. The significantly reduced layout of the MOOS scheme exhibits a great potential in high–resolution high–PPI displays, especially the next–generation VR/AR/MR displays.

## Figures and Tables

**Figure 1 micromachines-13-00839-f001:**
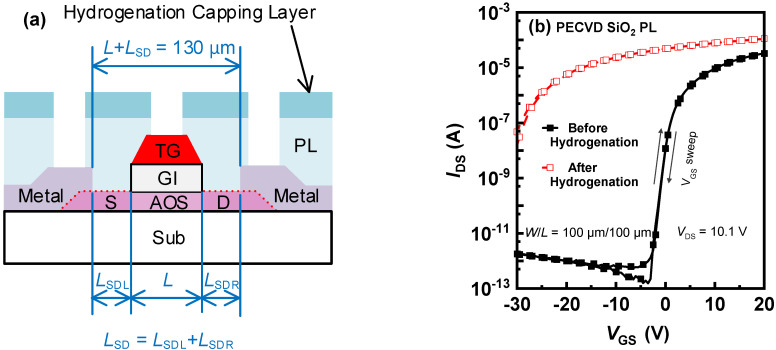
(**a**) The schematic cross–section of the SATG IGZO TFT with hydrogenation capping layer. Transfer characteristics of a–IGZO TFTs with (**b**) PECVD SiO_2_ passivation layer, (**c**) sputtered–AlO_X_ passivation layer, and (**d**) SiO_2_/Oxidized–AlO_X_ passivation layer before and after the hydrogenation.

**Figure 2 micromachines-13-00839-f002:**
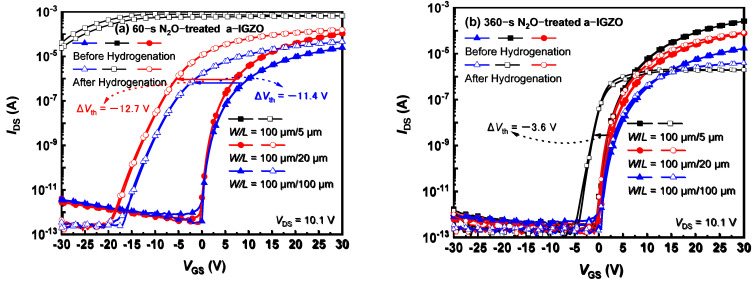
The transfer characteristics of TFTs with a–IGZO pretreated by N_2_O plasma for (**a**) 60 s and (**b**) 360 s before and after the hydrogenation.

**Figure 3 micromachines-13-00839-f003:**
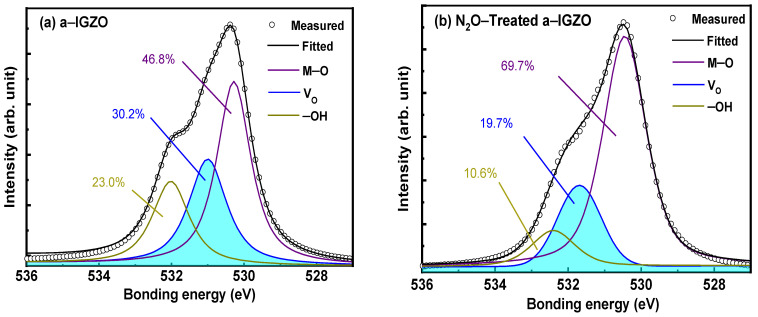
O1s XPS of (**a**) a–IGZO and (**b**) N_2_O–treated a–IGZO without the hydrogenation.

**Figure 4 micromachines-13-00839-f004:**
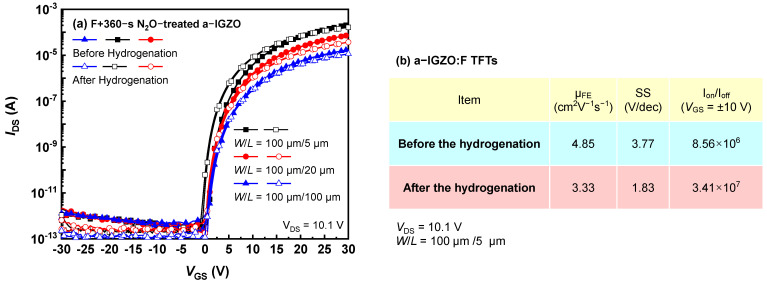
(**a**) The transfer characteristics of 360–s N_2_O–treated a–IGZO:F TFTs and (**b**) corresponding key parameters before and after the hydrogenation of the 5–μm–L a–IGZO:F TFTs.

**Figure 5 micromachines-13-00839-f005:**
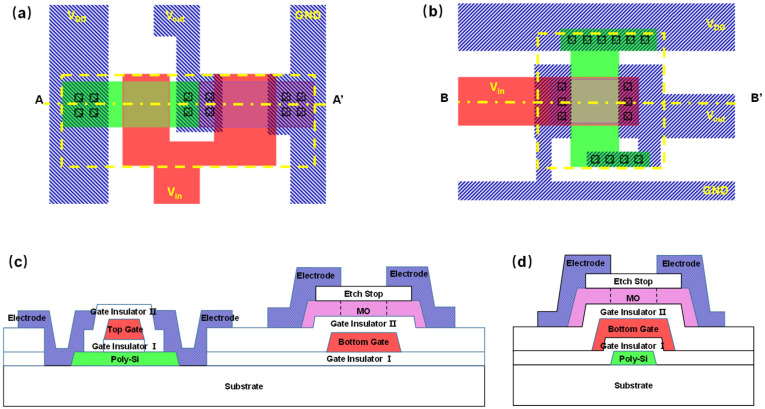
LTPO inverter layouts of the (**a**) side–by–side and (**b**) MOOS structures. The cross–sections along (**c**) A–A’ and (**d**) B–B’ directions.

**Figure 6 micromachines-13-00839-f006:**
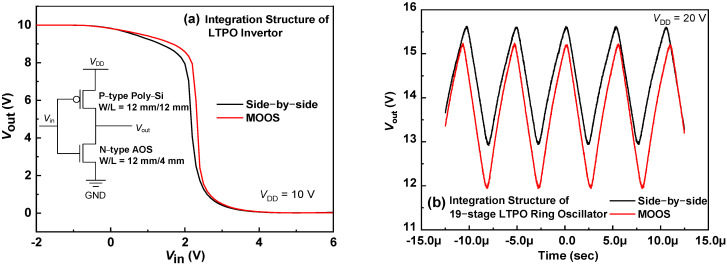
The comparison characteristics of LTPO Inverter (**a**) and 19–stage Ring Oscillators (**b**) in the Side–by–side & MOOS structures.

## Data Availability

Not applicable.

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
