# Peer review of "Compact Integration of Hydrogen–Resistant a–InGaZnO and Poly–Si Thin–Film Transistors"

_micromachines, 2022, doi:10.3390/mi13060839_

Round 1

Reviewer 1 Report

In the manuscript, the authors have proposed a more compact LTPO structure by stacking the H-resistant IGZO on poly-Si, and successfully verified using the basic circuit modules. The MOOS scheme exhibits a great potential for developing high-PPI displays. The manuscript is well organized and written, and the results and data are sufficient. Therefore, I recommend the publication of the study after the following issues are addressed.

1, After the hydrogenation, the resistance of IGZO:F SD was suggested to be lower than that of IGZO SD. Please give the specific resistance difference.

2, In Fig.3, the peak sum of these fitted subpeaks need to be included, directly verifying the fitting accuracy.

3, For the high mobility oxide TFTs, the authors should cite Nanomaterials, 2020, 10, 617; Nanomaterials, 2020, 10, 1782.

Reviewer 2 Report

Abstract needs light editing for English; make clear that aspects are novel to this paper. Think about breaking sentences up into shorter pieces so reader does not get lost.

Suggestion is to not use abbreviation 'PL', but spell out words for each usage -- it will be easier for comprehension.

Legend is Fig 2a is confused -- there is no magenta triangle curve, but rather blue non-filled triangle. The symbols should be redone to be more rationally assigned as in Fig 2b.

Data in Fig 2a is confusing because shift with length is not monotonic -- it would be better if there was more data to try to understand the trend.

Clarify in the description of Fig 3 that these samples have not seen the hydrogen anneal. 

The paper would be better if you could support your conjecture that the N2O treatment made the IGZO FTs more robust against a hydrogen treatment -- alternative explanation would be that it blocked hydrogen.

Figs 5 and 6b missing in the review copy of the paper. Thes figure seem essential for review.

The paper seems to have two unconnected parts -- (a) the hydrogen anneal studies on single IGZO FETs and (b) the stacked LPTO. The abstract gives the mistaken impression that (2) in some way enables the (1). You might consider breaking the work into two papers with (1) with its experimental data being the initial focus. 

While the paper seems well referenced concerning technical aspects, the text does no address why the motivation and what where it stands with respect to the state of the art. What is it's state of novelty? What does this enable?

Reviewer 3 Report

1.Why the Vth was negative shifted by H treatment?
2. The author should add some statment about the role of H in IGZO TFTs  for the introduction section (Journal of Alloys and Compounds 831 (2020) 154694);ACS Appl. Mater. Interfaces 2017, 9, 10798−10804。
3. Why the Vo is reduced by N2O treatment?
